# Plan before Solving: Problem-Aware Strategy Routing for Mathematical Reasoning with LLMs

## Abstract

Existing methods usually leverage a fixed strategy, such as natural language reasoning, code-augmented reasoning, tool-integrated reasoning, or ensemble-based reasoning, to guide Large Language Models (LLMs) to perform mathematical reasoning. Our analysis reveals that the single strategy cannot adapt to problem-specific requirements and thus overlooks the trade-off between effectiveness and efficiency. To address these issues, we propose Planning and Routing through Instance-Specific Modeling (PRISM), a novel framework that decouples mathematical reasoning into two stages: strategy planning and targeted execution. Specifically, we first curate a multi-strategy preference dataset, which we call `MathStrat`, capturing correctness, process quality, and computational efficiency for each problem–strategy pair. Then, we train a lightweight Strategy Adapter based on the dataset to obtain confidence distributions over the mentioned four reasoning strategies. At inference time, an adaptive routing policy dynamically tailors the reasoning approach based on predictor confidence. It directs the model to use single-strategy execution for high-confidence predictions, dual-strategy verification for competitive scenarios, or comprehensive multi-strategy exploration for uncertain cases. Extensive experiments across five mathematical reasoning benchmarks demonstrate that PRISM consistently outperforms individual strategies and ensemble baselines, achieving improvements ranging from 0.9% to 7.6% across different base models. The adaptive routing approach shows particularly strong benefits for mathematical reasoning tasks across diverse model architectures. Our code is released at `https://github.com/reml-group/PRISM`.

## 1 Introduction

Large Language Models (LLMs), such as ChatGPT and Qwen, have achieved significant advancements across diverse natural language processing tasks (Ma et al., 2025a;b). Notably, they have demonstrated strong performance in mathematical reasoning—a long-standing and challenging domain that requires precise logical inference, symbolic manipulation, and multi-step problem-solving (Gou et al., 2024b). Existing approaches to improving mathematical reasoning in LLMs can be broadly divided into three categories: (1) designing effective prompting methods for frozen LLMs (Trivedi et al., 2025), (2) developing strategies to enhance the capability of frozen LLMs (Didolkar et al., 2024), and (3) post-training LLMs on domain-specific data (Xia et al., 2025). Among these, the method of enhancing frozen LLMs through inference-time mechanisms such as chain-of-thought refinement (Wei et al., 2022b) and tool invocation (Xie et al., 2025) has garnered considerable attention due to its ease of deployment. Although these methods have achieved significant success, they still face two challenges.

**Challenge 1: One strategy does not fit all.** Existing methods primarily rely on isolated reasoning strategies, including Natural Language Reasoning (NLR) (Wang et al., 2024), Code-Augmented Reasoning (CAR) (Ye et al., 2024), Tool-Integrated Reasoning (TIR) (Li et al., 2024), and Ensemble-Based Reasoning (EBR) (Ranaldi et al., 2024), to enhance the mathematical reasoning of frozen LLMs. As illustrated in Figure 1, we evaluate the performance of these individual strategies in boosting the reasoning ability of Qwen-2.5-math across four question types sampled from MATH

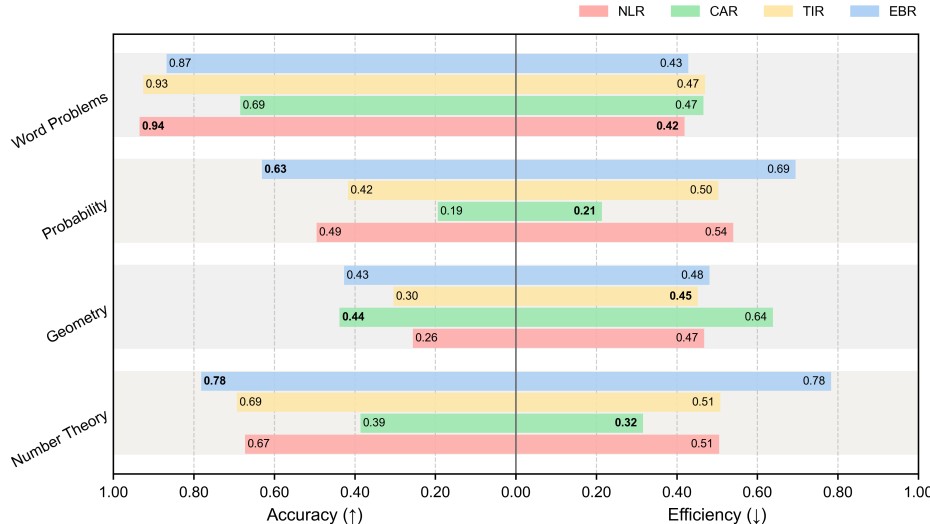

Figure 1: Performance of four reasoning strategies on four problem slices. Each slice (*e.g.*, Number Theory, Geometry) comprises over 100 instances drawn from the MATH, GSM8K, and SVAMP benchmarks.

(Hendrycks et al., 2021), GSM8K (Cobbe et al., 2021), and SVAMP (Patel et al., 2021). Our analysis reveals that no single strategy consistently outperforms others across diverse problem categories. This performance variance underscores a key limitation: rigidly adhering to a fixed reasoning paradigm fails to fully unlock the latent capabilities of frozen LLMs and hampers the adaptability to various problem types.

**Challenge 2: Trade-offs between efficiency and effectiveness are overlooked.** Current approaches (Xin et al., 2025; Zhang & Xiong, 2025) often disregard the computational cost, latency, and resource efficiency of reasoning strategies. As shown in Figure 1, we report the normalized inference efficiency of different strategies for enhancing the reasoning ability of Qwen2.5-7B across four question types. Notably, no single strategy consistently achieves the best efficiency. This observation suggests that a fixed reasoning paradigm leads to suboptimal deployments, where substantial computational expense does not yield commensurate improvements in accuracy.

To address these challenges, we introduce Planning and Routing through Instance-Specific Modeling (PRISM), a framework that decouples mathematical reasoning into two core stages: strategy planning and targeted execution. Specifically, we propose a data construction approach based on multi-strategy performance profiling, which systematically evaluates diverse reasoning strategies on each problem instance to generate fine-grained suitability distributions rather than single-strategy labels. To achieve this, we execute four distinct reasoning strategies (i.e., NLR, CAR, TIR, and EBR) on problems from standard benchmarks like MATH and GSM8k. Each resulting solution trajectory is then evaluated using a multi-faceted scoring function that considers correctness, process quality, and efficiency to generate per-strategy suitability scores. These raw scores are then transformed into a soft target distribution via a temperature-scaled softmax function. We then train a lightweight Strategy Adapter by minimizing the Kullback-Leibler (KL) divergence between its output and this target distribution, which encourages the model to capture the relative suitability of strategies for each instance. At inference time, the output from the predictor drives our problem-aware strategy routing that reconciles the efficiency-effectiveness trade-off through confidence-based execution paths: high-confidence predictions trigger streamlined single-path execution; competitive scores between strategies invoke dual-path verification for robustness; and diffuse uncertainty defaults to comprehensive multi-path exploration. Through this confidence-guided orchestration, PRISM achieves both strategic flexibility and computational efficiency, selecting the most suitable reasoning approach for each problem while scaling computational effort according to prediction certainty.

To verify the effectiveness and superiority, we evaluate PRISM across five standard mathematical reasoning benchmarks, including MATH500, GSM8K, AQUA-RAT (Ling et al., 2017), SVAMP, and ASDiv (Miao et al., 2020). Our experiments show that PRISM consistently delivers signifi-

cant performance gains. Notably, on the challenging MATH benchmark, our method achieves an accuracy of 53.2%, surpassing the best-performing single-strategy baseline (TIR) by 3.1% absolute. Furthermore, it outperforms the standard ensemble-based approach (EBR) by a margin of 2.5% absolute, demonstrating the superiority of pre-execution routing over post-hoc aggregation.

Our main contributions are as follows:

1. We introduce PRISM, a novel framework that decouples the mathematical reasoning process into two distinct stages: strategy planning and targeted execution. This is operationalized through a lightweight meta-predictor trained on `MathStrat`, our curated dataset of ~13,000 instances that provides rich, multi-faceted supervision signals that capture the relative suitability of various reasoning strategies.

2. We design a dynamic, verifier-free routing policy for inference. This policy interprets the predictor's output to adaptively select among single, dual, or multi-path execution modes, providing a principled mechanism to balance performance with computational cost.

3. We conduct extensive experiments across five standard mathematical benchmarks. The results demonstrate that PRISM consistently and significantly outperforms all single-strategy baselines and a standard ensemble method, validating the superiority of our problem-aware routing approach.

## 2 RELATED WORK

**Natural Language Reasoning** The dominant paradigm for complex reasoning in LLMs is Chain-of-Thought (CoT), which externalizes intermediate steps in natural language (Wei et al., 2022a). Recent efforts to improve NLR have focused on enhancing the quality of process data used for fine-tuning, with representative approaches such as bootstrapping via question back-translation (Yu et al., 2024; Lu et al., 2024b) and evolutionary rewriting of instructions (Luo et al., 2025). While effective for symbolic deduction, NLR's reliance on unstructured text makes it prone to arithmetic and logical errors in computationally intensive problems.

**Code-Augmented Reasoning** To address the computational limitations of NLR, CAR reframes mathematical problems as program generation tasks, offloading calculations to a deterministic code interpreter (Zhang et al., 2024). This is often implemented through prompting paradigms like Program-of-Thoughts (PoT) (Chen et al., 2023) or by fine-tuning models on interleaved text and code, as in Program-Aided Language Models (PAL) (Gao et al., 2023b). CAR excels at numerical precision but remains dependent on the initial natural language understanding for problem decomposition and program planning.

**Tool-Integrated Reasoning** TIR extends the role of LLMs from solvers to agents that dispatch tasks to external tools like calculators or symbolic solvers. This approach, exemplified by works such as ToRA (Gou et al., 2024a), creates a call-verify-iterate loop that enhances robustness on high-difficulty problems. Recent studies (Jin et al., 2024; Wu et al., 2024) have also demonstrated strong performance by leveraging advanced integrated environments like the GPT-4 Code Interpreter for complex problem-solving (Nguyen & Allan, 2024). The primary trade-off for TIR is the increased complexity and latency associated with tool selection and orchestration.

**Ensemble-Based Reasoning** To improve robustness with minimal engineering, lightweight ensemble methods aggregate multiple solution trajectories. The most prominent example is Self-Consistency, which samples multiple CoT paths and selects the answer via majority voting (Wang et al., 2023). Other works extend this by exploring more complex reasoning structures like a Tree of Thoughts (Yao et al., 2023b). These methods (Zhang et al., 2025; Yao et al., 2023a; Xia et al., 2025), however, are fundamentally forms of post-hoc selection, requiring the generation of multiple costly trajectories before aggregation and failing to identify the optimal strategy in advance. Related work has also explored selecting among reasoning modes before inference, such as routing between CoT and PAL using an external selector and assigning problems to predefined reasoning types via supervised classification. (Yue et al., 2023);(Zhao et al., 2023)

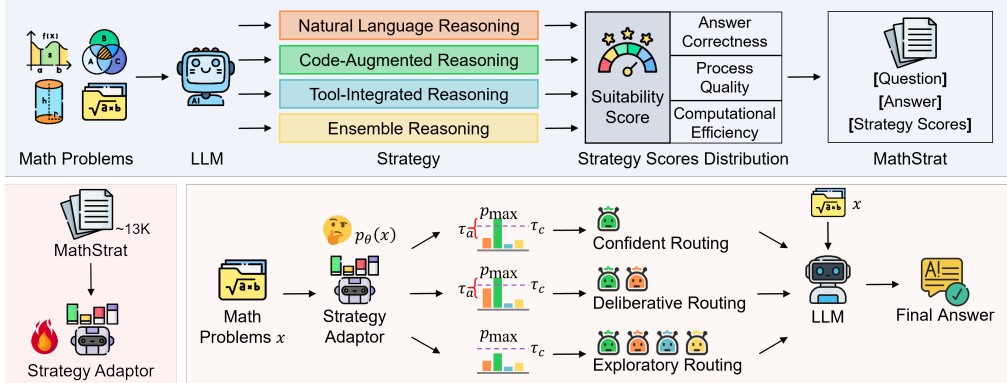

Figure 2: Overview of the PRISM framework. The framework consists of two stages: Offline training (top and bottom-left), where mathematical problems are solved under multiple reasoning strategies and build a dataset called `MathStrat` for training the Strategy Adapter; Online inference (bottom-right), where the Strategy Adapter guides adaptive routing to produce the final answer.

## 3 Methodology

The PRISM framework is designed in two stages that decouple strategy planning from execution. The first stage involves an offline training of a Strategy Adapter (SA). This model learns to map a given problem instance to a suitability distribution over a set of reasoning strategies. The second stage is the online inference, where the prediction of SA guides an adaptive routing policy to select an execution pathway dynamically. The subsequent sections detail the mentioned stages: Section 3.1 describes the formulation and training of the SA, and Section 3.2 presents the adaptive routing policy used at inference.

### 3.1 Strategy Adapter

As outlined in Section 1, existing reasoning strategies can be broadly categorized into four paradigms: NLR, CAR, TIR, and EBR. To achieve dynamic strategy selection, we propose the strategy adaptation pipeline, implemented in two stages: (1) collection of strategy preference data, and (2) training of the strategy suitability assessment model.

**Collection of Strategy Preference Data.** To generate effective training signals for strategy selection, we evaluate each approach across multiple performance aspects. We construct supervision signals using three complementary dimensions that capture the essential trade-offs: (1) answer correctness, which determines whether the strategy produces the correct solution to the given mathematical problem; (2) process quality, which evaluates whether the strategy follows mathematically valid reasoning and avoids logical errors or redundant operations; and (3) computational efficiency, which measures whether the strategy achieves results within reasonable time and resource consumption. We quantify computational efficiency using two metrics: the actual wall-clock inference time and the number of generated tokens, both normalized via within-instance min–max scaling. To implement this evaluation framework, we execute all four reasoning strategies (NLR, CAR, TIR, and EBR) on each problem instance $x$ using identical base models and decoding configurations. This yields three measurements per strategy $s$: (i) binary correctness $corr(s, x) \in \{0, 1\}$ indicating successful problem solving; (ii) process quality $qual(s, x) \in [0, 1]$ from an automated evaluator that penalizes invalid steps and redundant reasoning (Xia et al., 2025); and (iii) efficiency score $eff(s, x) \in [0, 1]$ computed from raw timing and output length metrics. Specifically, we define the efficiency score $eff(s, x)$ as:

$$eff(s, x) = 1 - \tfrac{1}{2}\big(\hat{t}(s, x) + \hat{\ell}(s, x)\big), \tag{1}$$

where the normalized timing and length components $\hat{t}(s, x)$ and $\hat{\ell}(s, x)$ are obtained through min-max scaling within each problem instance:

$$\hat{t}(s, x) = \frac{t(s, x) - \min_{s'} t(s', x)}{\max_{s'} t(s', x) - \min_{s'} t(s', x) + \epsilon}, \tag{2}$$

$$\hat{\ell}(s, x) = \frac{\ell(s, x) - \min_{s'} \ell(s', x)}{\max_{s'} \ell(s', x) - \min_{s'} \ell(s', x) + \epsilon}, \tag{3}$$

and $\epsilon > 0$ ensures numerical stability. The three signals are aggregated using fixed weights $(w_C, w_Q, w_U)$ to yield a per-strategy suitability score $score(s, x)$:

$$score(s, x) = w_C \cdot corr(s, x) + w_Q \cdot qual(s, x) + w_U \cdot eff(s, x). \tag{4}$$

We then form a soft supervision target distribution $\mathbf{y}(x)$ by applying a temperature-scaled softmax function over the four scores within the same instance, where $y(s, x)$ represents the target probability for strategy $s$ on instance $x$:

$$y(s, x) = \frac{\exp\big(score(s, x)/\tau\big)}{\sum_{s' \in \mathcal{S}} \exp\big(score(s', x)/\tau\big)}, \qquad \tau = 0.5. \tag{5}$$

**Strategy Adapter Training.** We train the Strategy Adapter $f_\theta$ to output logits $z_\theta(x) \in \mathbb{R}^{|\mathcal{S}|}$ and predict a probability distribution $p_\theta(x) = \mathrm{softmax}\big(z_\theta(x)\big)$ over strategies for each problem instance. Our training objective is designed to match the full target distribution while explicitly stabilizing the ranking of the top strategy. The primary objective minimizes the Kullback–Leibler (KL) divergence between the target distribution $\mathbf{y}(x)$ and the predicted distribution $p_\theta(x)$:

$$\mathcal{L}_{\mathrm{dist}}(\theta) = \frac{1}{N} \sum_{i=1}^{N} \mathrm{KL}(\mathbf{y}(x_i) \,\|\, \mathbf{p}_\theta(x_i)). \tag{6}$$

To reinforce learning of the top-ranked strategy, we add an auxiliary cross-entropy loss. Let $s_i^* = \arg\max_{s \in \mathcal{S}} S(s, x_i)$ be the best strategy for instance $x_i$. The auxiliary loss is:

$$\mathcal{L}_{\mathrm{ord}}(\theta) = -\frac{1}{N} \sum_{i=1}^{N} \log p_{\theta, s_i^*}(x_i). \tag{7}$$

This auxiliary loss explicitly enforces top-1 selection, which is essential for our routing mechanism, and it also prevents the predictor from producing overly soft distributions when trained with the KL term alone. The final loss combines these objectives:

$$\mathcal{L}(\theta) = \mathcal{L}_{\mathrm{dist}}(\theta) + \lambda \mathcal{L}_{\mathrm{ord}}(\theta), \tag{8}$$

where $\lambda$ is a hyperparameter that balances the two objectives. This combined distributional and ranking-aware objective encourages the model to capture the relative suitability of strategy families, which guides the inference-time routing policy. We implement the Strategy Adapter as a lightweight language model (e.g., 1.5B parameters) trained on our curated dataset of approximately 13,000 problem instances with multi-strategy performance evaluations. Upon completion of training, the adapter demonstrates effective suitability assessment capabilities, with a representative example of its prediction behavior provided in Appendix A.6.

## 3.2 Adaptive Routing Policy at Inference

The Strategy Adapter produces a probability distribution $p_\theta(x)$ over strategies for each problem instance $x$. Simply selecting the highest-probability strategy, however, would result in uniform single-strategy execution regardless of prediction confidence (as illustrated in Figure 1). This approach fails to exploit opportunities for computational efficiency when predictions are highly confident or for enhanced robustness when predictions are uncertain.

Our adaptive routing policy interprets the predictor output through a confidence-based framework that dynamically selects among three execution modes: *Confident*, *Deliberative*, and *Exploratory* routing. The mode selection depends on two calibrated thresholds: a confidence threshold $\tau_c$ and an ambiguity margin $\tau_a$, which are optimized through grid search on a validation set (see Appendix A.4 for details), applied to the top two predicted probabilities $p_{\max}$ and $p_{\mathrm{2nd}}$. *Confident*

*Routing* ($p_{\max} \geq \tau_c$ and $(p_{\max} - p_{2nd}) \geq \tau_a$) executes only the single best-ranked strategy when the predictor exhibits high confidence with a clear preference. *Deliberative Routing* ($p_{\max} \geq \tau_c$ and $(p_{\max} - p_{2nd}) < \tau_a$) executes the top two strategies when confidence is high but rankings are close, selecting the answer with agreement or, in case of disagreement, the answer from the higher-confidence strategy. *Exploratory Routing* ($p_{\max} < \tau_c$) executes all available strategies when predictor confidence is insufficient, again using majority voting for final answer selection. The routing mode distribution under different threshold configurations on this validation set is analyzed in Figure 4 (left), demonstrating that lower confidence thresholds lead to increased confident routing (single-strategy execution) while higher thresholds favor more exploratory multi-strategy approaches. For multi-strategy modes, answers undergo standardization to normalize numerical formats before voting, with ties resolved by selecting the strategy with highest predicted probability. This mechanism provides principled computational resource allocation without requiring external verification components.

## 4 EXPERIMENT

### 4.1 SETUP

**Datasets and Baselines**  We use as diverse mathematical datasets as possible for experiments. In addition to the widely used MATH (Hendrycks et al., 2021) and GSM8K (Cobbe et al., 2021) dataset, we also adopt AQUA-RAT (Ling et al., 2017), SVAMP (Patel et al., 2021) and ASDiv (Miao et al., 2020). These datasets cover multiple fields of mathematics, such as elementary arithmetic problems, mathematical algebra, inferential counting, and probability number theory. They also span a wide range of difficulty levels, including simple elementary school math problems, intermediate-level questions, and even Olympiad-style competition problems. We select large language models from three series. Our experiments use Qwen2.5-Math-7B (Yang et al., 2024), Deepseek-math-7b-v1 (Lu et al., 2024a), and Llama-3-8B to conduct thorough evaluations. For evaluation metrics, we report **Pass@k** accuracy, where a problem is considered solved if the correct answer appears in the top-$k$ generated solutions. Specifically, Pass@1 reflects single-shot correctness, while Pass@5 captures the probability of producing at least one correct solution among five independent generations.

**Reasoning Approaches**  As mentioned in previous studies, our experiment also involves four other mathematical reasoning approaches like CoT (Wei et al., 2022a), PAL (Gao et al., 2023a), ToRA (Zhang et al., 2023), Hybrid (Yue et al., 2023). Chain-of-Thought (CoT) prompting is a technique designed to elicit more robust reasoning from LLMs by encouraging them to generate a series of intermediate, step-by-step rationales before concluding with a final answer. Program-Aided Language Models (PAL) introduce a neuro-symbolic approach that offloads the reasoning and calculation logic to an external tool. Tool-Augmented Reasoning Agent (ToRA) can interleave natural language reasoning steps with calls to different tools, such as a calculator, a symbolic solver, or retrieval APIs. Hybrid approaches aim to combine the strengths of different reasoning paradigms to achieve superior performance and robustness.

### 4.2 MAIN RESULTS

Table 1 shows performance across three base models and five mathematical reasoning benchmarks. PRISM achieves average improvements of 0.9% on Qwen2.5-Math-7B, 2.9% on Deepseek-math-7b-v1, and 7.6% on Llama-3-8B over the best single strategies. The inverse relationship between relative improvement and base model capability suggests that strategic routing provides greater value when addressing model limitations. The results confirm our central observation that no single strategy dominates across all benchmarks—while ToRA excels on MATH500, PAL leads on GSM8K for Qwen (95.3%), and performance varies dramatically across datasets. PRISM effectively handles this heterogeneity through adaptive strategy selection. The method substantially outperforms the Hybrid baseline, particularly on complex reasoning tasks like AQUA-RAT, demonstrating that pre-execution routing is more effective than post-hoc strategy aggregation. Individual strategies exhibit high variance (PAL ranges from 13.5% to 95.3%), while PRISM maintains consistent performance across all test conditions.

Table 1: Performance comparison of different mathematical reasoning strategies across three base language models and five datasets. CoT refers to Chain-of-Thought reasoning, PAL to Program-Aided Language models, ToRA to Tool-integrated Reasoning Agent, and Hybrid to ensemble-based approaches. PRISM represents our proposed adaptive routing framework. Pass@k denotes the percentage of problems for which at least one correct solution appears in the top-k generated outputs. Blue highlighting indicates the best performance for each model-dataset combination.

| Model | Approach | MATH500 | | GSM8K | | AQUA-RAT | | SVAMP | | ASDiv | | Average | |
|---|---|---|---|---|---|---|---|---|---|---|---|---|---|
| | | Pass@1 | Pass@5 | Pass@1 | Pass@5 | Pass@1 | Pass@5 | Pass@1 | Pass@5 | Pass@1 | Pass@5 | Pass@1 | Pass@5 |
| Qwen2.5-Math-7B | CoT | 21.2 | 50.8 | 78.1 | 93.5 | 37.0 | 57.5 | 85.7 | 94.6 | 82.8 | 92.5 | 61.0 | 77.8 |
| | PAL | 30.4 | 55.4 | 84.8 | 95.3 | 18.1 | 44.1 | 13.5 | 45.2 | 86.3 | 93.7 | 46.6 | 66.7 |
| | ToRA | 41.4 | 62.0 | 69.4 | 94.5 | 47.2 | 64.6 | 75.8 | 96.2 | 75.7 | 93.6 | 61.9 | 82.2 |
| | Hybrid | 37.2 | 53.2 | 24.0 | 68.2 | 15.4 | 44.5 | 21.8 | 63.9 | 15.9 | 50.1 | 22.9 | 56.0 |
| | PRISM (ours) | 46.2 | 64.4 | 86.7 | 96.0 | 42.1 | 64.8 | 91.8 | 96.9 | 86.5 | 93.6 | 70.7 | 83.1 |
| Deepseek-math-7b-v1 | CoT | 43.0 | 57.2 | 87.6 | 93.3 | 33.5 | 56.7 | 83.7 | 92.8 | 85.6 | 91.2 | 68.6 | 79.2 |
| | PAL | 38.0 | 53.2 | 83.9 | 91.8 | 47.6 | 55.1 | 84.8 | 90.2 | 84.2 | 89.7 | 67.7 | 76.0 |
| | ToRA | 32.2 | 49.2 | 78.3 | 93.0 | 38.6 | 58.7 | 76.5 | 92.4 | 79.7 | 91.5 | 61.1 | 77.0 |
| | Hybrid | 12.6 | 30.8 | 60.0 | 90.1 | 26.3 | 50.9 | 71.9 | 93.2 | 68.2 | 90.2 | 47.8 | 70.9 |
| | PRISM (ours) | 52.2 | 61.6 | 87.8 | 93.6 | 45.3 | 62.2 | 88.0 | 94.3 | 88.6 | 92.3 | 72.4 | 79.9 |
| Llama-3-8B | CoT | 13.6 | 29.8 | 45.6 | 76.8 | 17.7 | 36.6 | 64.6 | 88.6 | 22.6 | 60.1 | 32.8 | 58.4 |
| | PAL | 10.4 | 22.9 | 54.3 | 78.4 | 16.5 | 35.4 | 72.4 | 89.3 | 13.5 | 31.7 | 33.4 | 51.5 |
| | ToRA | 11.0 | 25.0 | 43.9 | 75.1 | 14.6 | 33.1 | 65.6 | 89.2 | 21.9 | 60.0 | 31.4 | 56.5 |
| | Hybrid | 11.8 | 26.2 | 44.6 | 88.9 | 10.9 | 37.7 | 22.0 | 62.7 | 25.1 | 62.7 | 22.9 | 55.6 |
| | PRISM (ours) | 15.2 | 36.2 | 53.0 | 78.5 | 14.6 | 33.1 | 66.1 | 89.6 | 63.7 | 83.1 | 42.5 | 64.1 |

## 4.3 ABLATION STUDY ON ROUTING COMPONENTS

To understand the contribution of each routing component, we conducted progressive ablation experiments across GSM8K, MATH500, and Hungarian Math datasets. As detailed in 2, adding confident routing shows mixed results—76.0% on GSM8K (below the 78.1% CoT baseline) but improvements on MATH500 (28.6% vs 21.2%) and Hungarian Math (50.0% vs 40.6%). Incorporating deliberative routing provides continued gains on MATH500 (32.2%) with variable performance elsewhere. The complete PRISM system achieves substantial improvements across all datasets (86.7%, 46.2%, and 53.1% respectively), with dramatic jumps from the previous configuration demonstrating that the full adaptive routing policy is essential for optimal performance. These results validate that individual routing modes provide limited benefits, while the intelligent coordination of all components through problem-aware strategy selection delivers significant performance gains.

Table 2: Ablation study of adaptive routing components using the Qwen2.5-Math-7B model.

| Setting | GSM8K | | MATH500 | | Hungarian Math | |
|---|---|---|---|---|---|---|
| | pass@1 | pass@5 | pass@1 | pass@5 | pass@1 | pass@5 |
| CoT baseline | 78.1 | 93.5 | 21.2 | 50.8 | 40.6 | 56.3 |
| Confident | 76.0 | 95.0 | 28.6 | 56.1 | 50.0 | 62.5 |
| Confident + Deliberative | 75.8 | 95.6 | 32.2 | 57.3 | 43.8 | 50.0 |
| PRISM | **86.7** | **96.0** | **46.2** | **64.4** | **53.1** | **71.9** |

## 4.4 PERFORMANCE-EFFICIENCY TRADE-OFF

To examine the computational efficiency of our adaptive routing approach, we measured performance and resource consumption across different framework configurations, as shown in Figure 3. The baseline CoT approach achieves 9.5% Pass@1 accuracy with 5,200ms inference time and 45-token average output length. Adding the Strategy Adapter with confident routing (SA+Conf.) shows minimal performance improvement to 10.0% but increases computational cost to 6,300ms and 50 tokens. The combination of confident and deliberative routing (SA+Conf.+Delib.) achieves 17.8% accuracy while maintaining similar efficiency profiles at 6000ms and 80 tokens. The complete PRISM system demonstrates substantial performance gains, reaching 33.3% Pass@1 accuracy while achieving better efficiency than intermediate configurations at 5,600ms inference time and 85 tokens output length. This efficiency-performance trade-off reveals that the full adaptive routing policy not

only improves accuracy but also optimizes resource utilization by intelligently selecting execution pathways. The results indicate that the Strategy Adapter alone provides limited benefits, but when combined with the complete adaptive routing mechanism, it enables significant performance improvements while maintaining computational efficiency. This validates our design choice of integrating prediction-guided strategy selection with dynamic execution pathways rather than relying on individual components in isolation.

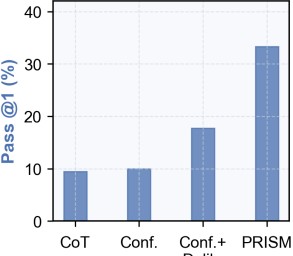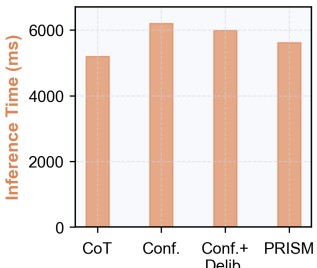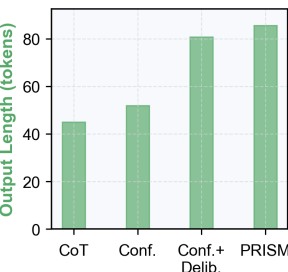

Figure 3: Analysis of PRISM framework components across three key metrics. (a) Pass@1 accuracy shows substantial gains with the full system. (b) Inference time and (c) output length demonstrate that PRISM achieves higher performance with better or comparable computational efficiency than intermediate configurations.

## 4.5 SCALABILITY ANALYSIS

To evaluate the scalability of our approach, we conducted experiments across Qwen2.5 models ranging from 1.5B to 72B parameters on GSM8K and MATH500 benchmarks, using Qwen2.5 7B with chain-of-thought prompting as our baseline. As illustrated in Figure 4 (right), PRISM demonstrates consistent improvements over the baseline across all model scales, achieving accuracy from 74.2% to 95.2% on GSM8K and 32.3% to 78.4% on MATH500. The framework exhibits distinct scaling patterns across benchmarks: steady improvements on GSM8K with notable gains from 7B to 32B parameters, and more dramatic scaling effects on MATH500 where performance nearly doubles from smallest to largest models. These scaling results validate that adaptive strategy selection provides robust benefits across different model capacities. The sustained improvements across parameter scales demonstrate that the framework generalizes effectively and does not depend on specific model characteristics to achieve performance gains. Importantly, since PRISM operates as a training-free approach that works purely through inference-time strategy selection, it can be readily applied to any pre-trained model without requiring additional fine-tuning or domain-specific training, making it broadly applicable across different model families and computational budgets.

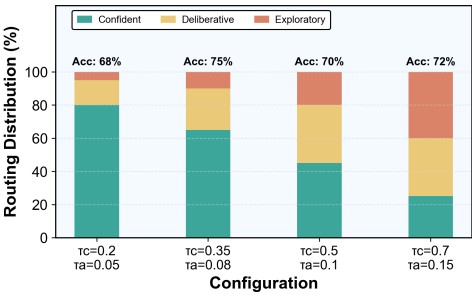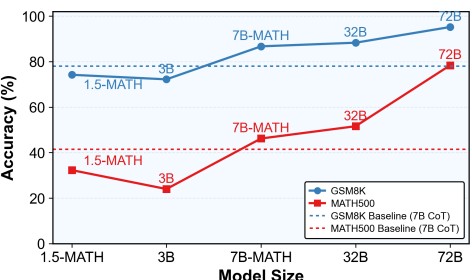

Figure 4: Left: Routing mode distribution analysis across different confidence threshold configurations. The stacked bars show the percentage of problems routed to each execution mode (Confident, Deliberative, Exploratory) for varying $\tau_c$ values while keeping $\tau_a = 0.08$. Right: Scalability of PRISM across Qwen2.5 models of varying sizes on GSM8K and MATH500 benchmarks. Dotted lines indicate the baseline performance of Qwen2.5-7B with standard chain-of-thought prompting.

## 4.6 STRATEGY ADAPTER BEHAVIOR ANALYSIS

We analyze the prediction behavior patterns of our Strategy Adapter across different mathematical reasoning datasets to validate its learned strategy selection characteristics. This analysis ex-

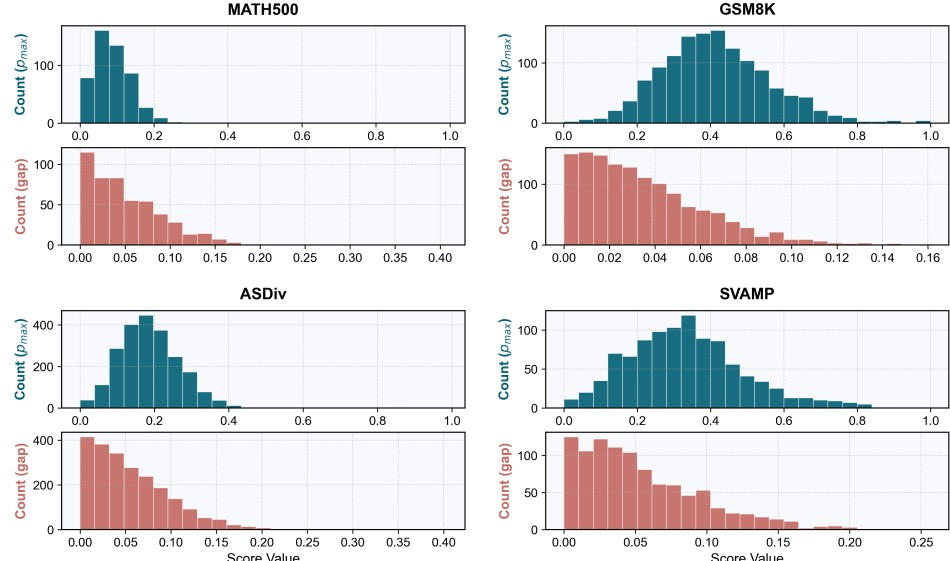

Figure 5: Strategy Adapter behavior across mathematical reasoning datasets. Top row shows prediction confidence ($p_{\max}$) distributions, bottom row shows strategy competition gaps ($p_{\max} - p_{\text{2nd}}$). The SA exhibits dataset-appropriate confidence levels: conservative predictions on competition problems (MATH500) and higher confidence on elementary problems (ASDiv, SVAMP), while maintaining competitive strategy landscapes across all datasets.

amines both the confidence levels in predictions and the competitive landscape among strategies. Figure 5 presents the distributions of prediction confidence ($p_{\max}$) and strategy competition gaps ($p_{\max} - p_{\text{2nd}}$) across four mathematical reasoning datasets. The results reveal several important patterns that validate our framework design. First, the SA exhibits prediction confidence patterns that correlate with dataset complexity. On MATH500, which contains competition-level mathematical problems, the predictor shows notably conservative behavior with prediction confidence ($p_{\max}$) concentrated in the 0.0 to 0.2 range. This low confidence reflects both the inherent difficulty of these problems and the SA's learned caution when dealing with competition-level mathematics, where strategy effectiveness is less predictable.

In contrast, datasets containing more elementary mathematical problems show progressively higher prediction confidence. GSM8K demonstrates moderate confidence levels with $p_{\max}$ distributed across $0.2 \sim 0.6$, while ASDiv and SVAMP exhibit relatively higher confidence with peaks around $0.3 \sim 0.4$. This graduated confidence pattern indicates that the SA has successfully learned to associate problem complexity with prediction uncertainty, demonstrating sophisticated meta-reasoning about strategy applicability. The adaptive confidence calibration also suggests that the SA effectively captures the inherent variability in strategy effectiveness across different mathematical domains, with higher uncertainty appropriately assigned to problems where multiple strategies might yield similar performance. Furthermore, the distribution shapes themselves provide insight into the SA's decision-making process: sharp peaks indicate clear strategy preferences for certain problem types, while flatter distributions suggest scenarios where multiple strategies remain viable options. The strategy competition analysis reveals consistently small gaps between top strategies across all datasets. This competitive landscape validates our adaptive routing design rationale: the narrow margins between strategy preferences require nuanced confidence-based decision making rather than simple winner-take-all selection. Additionally, we provide a detailed analysis of strategy performance patterns and inter-strategy correlations across datasets in Appendix A.5.

### 4.7 Problem Difficulty and Subject Analysis

To understand how problem characteristics influence strategy effectiveness and routing decisions, we conduct a fine-grained analysis on MATH500 across two dimensions: difficulty levels (1-5) and mathematical subjects (Precalculus, Counting & Probability, Geometry, Algebra, Intermediate Algebra, Number Theory, Prealgebra). Figure 6 presents the performance breakdown and routing behavior patterns.

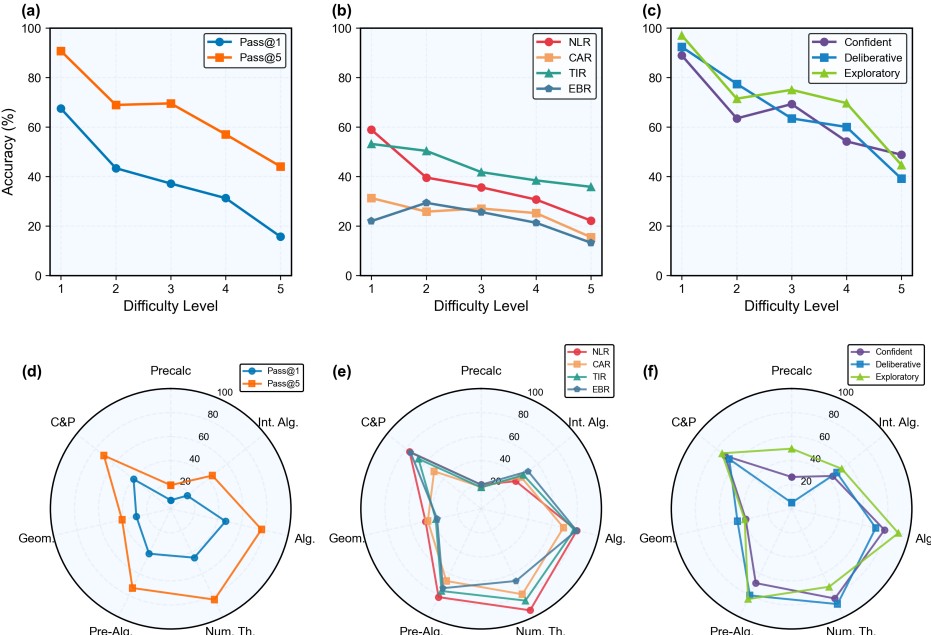

Figure 6: Problem difficulty and subject analysis on MATH500. (a-c) Strategy performance and routing behavior across five difficulty levels. (d-f) Subject-specific strategy suitability and routing mode distributions across seven mathematical domains.

**Difficulty-stratified performance.** The inter-strategy performance gap peaks at moderate difficulty (Levels 3-4), with approximately 30% separation between TIR (45%) and CAR (15%). This pattern reflects fundamental discrimination dynamics: easy problems (Level 1-2) offer limited discrimination as all strategies succeed, while hard problems (Level 5) reduce discrimination from a low baseline floor where all strategies struggle. The moderate difficulty range (Level 3-4) creates the optimal regime for adaptive routing, where problems are neither trivial nor universally intractable. Correspondingly, PRISM's routing mode distribution adapts intelligently: confident routing dominates at Level 1-2 (90%+), while the system progressively shifts toward balanced multi-strategy exploration at Level 5, demonstrating learned uncertainty calibration.

**Subject-specific strategy suitability.** Performance across mathematical subjects reveals distinct strategy-domain affinities. TIR consistently dominates in symbolic manipulation domains (Precalculus, Algebra, Intermediate Algebra) with 45-50% accuracy, while all strategies exhibit comparable struggles with probabilistic reasoning (Counting & Probability: 10-15%). Geometry triggers maximum routing diversity, with exploratory mode reaching 50%—indicating the Strategy Adapter's recognition of high inter-problem variability within this subject. The Pass@5 benefit also varies by subject: Number Theory shows +29% absolute gain over Pass@1, suggesting that sampling diversity provides greater value in domains with high solution path variability.

The difficulty-subject interaction reveals PRISM's dual-dimensional adaptivity: routing decisions respond to both problem complexity and domain characteristics, delivering maximum benefit where strategy differentiation is most pronounced.

## 5 CONCLUSION

We introduce a problem-aware strategy routing framework termed PRISM, which decouples mathematical reasoning into strategy planning and targeted execution. Specifically, it leverages a multi-strategy performance profiling mechanism to curate a 13K strategy preference dataset `MathStrat`. Then, a strategy adaptor is trained on this dataset to perform policy routing for the given problem at inference time. Extensive experiments with three language models across five datasets, combined with comprehensive ablation studies and efficiency analysis, demonstrate the effectiveness, superiority, and scalability of PRISM.

## ETHICAL STATEMENT

This study strictly adheres to academic integrity standards. We confirm that all research work is original and does not contain any form of plagiarism or data falsification. No personal privacy information is involved in the research process. The aim of this study is to promote positive development in the field of mathematical reasoning of large language model. We have carefully evaluated its potential social impact and are confident that it will not bring direct negative ethical risks. All authors have made substantial contributions to the research results and agree to the final submission of the manuscript.

## REPRODUCIBLE STATEMENT

To ensure the reproducibility of this study, we provide all necessary code, data, and experimental configuration details. Code: All the implementation code, model scripts, and experimental procedures of this study have been open sourced on this website `https://anonymous.4open.science/r/PRISM-1EAE/`. The code repository includes detailed README.md files to guide environment configuration and code execution. Dataset: The core dataset used in this study is publicly available as an open-source resource and can be readily accessed for research purposes.

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

# A  APPENDIX

## A.1  ADAPTIVE STRATEGY ROUTING ALGORITHM

This section provides the detailed pseudo-code for the PRISM adaptive routing policy, as referenced in Section 3.2.

---

**Algorithm 1** ADAPTIVE ROUTING POLICY AT INFERENCE

---

**Require:** Problem $P$; Preference Model $M$; Set of $k$ reasoning strategies $\mathcal{S} = \{\sigma_1, \sigma_2, \ldots, \sigma_k\}$; Confidence threshold $\tau_c$; Ambiguity threshold $\tau_a$

**Ensure:** Final answer $A$; Used routing mode $R$; Set of executed strategies $\Sigma^*$

1: $\mathbf{p} \leftarrow M(P)$          $\triangleright$ Predict strategy probabilities $p(\sigma_i \mid P)$ for all $\sigma_i \in \mathcal{S}$

2: $i_{\max} \leftarrow \arg\max_i \mathbf{p}_i$;   $p_{\max} \leftarrow \mathbf{p}_{i_{\max}}$;   $\sigma_{\max} \leftarrow \sigma_{i_{\max}}$

3: $i_{\text{2nd}} \leftarrow \arg\max_{i \neq i_{\max}} \mathbf{p}_i$;   $p_{\text{2nd}} \leftarrow \mathbf{p}_{i_{\text{2nd}}}$;   $\sigma_{\text{2nd}} \leftarrow \sigma_{i_{\text{2nd}}}$

4:          $\triangleright$ Route based on the predicted probability distribution

5: **if** $p_{\max} \geq \tau_c \land (p_{\max} - p_{\text{2nd}}) \geq \tau_a$ **then**

6:     $R \leftarrow$ CONFIDENT          $\triangleright$ **Confident Routing**

7:     $\Sigma^* \leftarrow \{\sigma_{\max}\}$

8:     $A \leftarrow \sigma_{\max}(P)$

9: **else if** $p_{\max} \geq \tau_c \land (p_{\max} - p_{\text{2nd}}) < \tau_a$ **then**

10:     $R \leftarrow$ DELIBERATIVE          $\triangleright$ **Deliberative Routing**

11:     $\Sigma^* \leftarrow \{\sigma_{\max}, \sigma_{\text{2nd}}\}$

12:     $A_1 \leftarrow \sigma_{\max}(P)$

13:     $A_2 \leftarrow \sigma_{\text{2nd}}(P)$

14:     $A \leftarrow \text{Vote}\left(\{A_1, A_2\}\right)$

15: **else**

16:     $R \leftarrow$ EXPLORATORY          $\triangleright$ **Exploratory Routing**

17:     $\Sigma^* \leftarrow \mathcal{S}$

18:     $\mathcal{A} \leftarrow \emptyset$

19:     **for** $\sigma_i \in \mathcal{S}$ **do**

20:        $A_i \leftarrow \sigma_i(P)$

21:        $\mathcal{A} \leftarrow \mathcal{A} \cup \{A_i\}$

22:     **end for**

23:     $A \leftarrow \text{Vote}(\mathcal{A})$

24: **end if**

25: **return** $(A, R, \Sigma^*)$

---

## A.2 Preference Data Example

Figure 7 presents a representative case of our multi-strategy performance evaluation process for collecting training data. The case shows a trigonometric function analysis problem where we execute all four reasoning strategies and collect comprehensive performance metrics.

The data collection captures three complementary dimensions as described in our methodology: answer correctness (validity scores), process quality (redundancy measures), and computational efficiency (execution time and output length). As illustrated in this case, different strategies exhibit distinct performance profiles: NLR achieves moderate validity (0.23) but suffers from high redundancy (0.68), while TIR demonstrates better process quality with higher validity (0.29) and lower redundancy (0.72). CAR shows the poorest validity (0.18) with the highest redundancy (0.73), and EBR balances the highest validity (0.35) with acceptable redundancy (0.66).

The condensed output summaries reveal how each strategy approaches the same trigonometric problem differently: NLR relies on numerical approximation methods, CAR attempts algebraic simplification, TIR uses analytical identities, and EBR combines multiple approaches. This diversity in solution paths, combined with varying performance across the three evaluation dimensions, provides rich supervision signals that enable our Strategy Adapter to learn nuanced associations between problem characteristics and strategy effectiveness. The multi-faceted evaluation ensures that our training data captures the essential trade-offs between correctness, reasoning quality, and computational cost.

**Example of Collected Preference Data**

| Strategy | Time | Length | Validity | Redundancy | Condensed Output Summaries |
|---|---|---|---|---|---|
| NLR | 7.6281 | 2151 | 0.23 | 0.68 | 1. The equation $\frac{x}{100}$=sinx is analyzed by comparing the linear function f(x)=x/100 and the periodic function g(x)=sinx. 
 2. Since $|\sin x|\le 1|\sin x|\le 1$, intersections occur only for $|x|\le 100$, and within this range, the approximately 31.83 periods of sinx suggest about 64 intersections (two per period plus endpoints). 
 3. A Python script numerically checks for sign changes between the functions over the interval [−100,100] using a high-resolution grid of 10,000 points. 
 4. The numerical computation yields 63 intersections, correcting the initial analytical estimate and providing the final solution count. |
| CAR | 12.0192 | 3197 | 0.18 | 0.73 | 1. The function is first simplified using trigonometric identities to the core form f(x)=√2sin(2x+$\frac{\pi}{4}$) for easier analysis. 
 2. The period is calculated directly from the simplified function's angular frequency, yielding a smallest positive period of π. 
 3. By analyzing the sine function over the transformed interval of its argument, the maximum of sin(u)sin(u) is found to be 1 and the minimum to be $-\frac{\sqrt{2}}{2}$ 
 4. The extreme values are scaled by the amplitude √2 to give the final results: a maximum of √2 and a minimum of −1 for f(x) |
| TIR | 7.0319 | 1907 | 0.29 | 0.72 | 1. The original trigonometric expression is simplified using sum-to-product and double-angle identities to obtain f(x)=sin2x+cos2x. 
 2. The simplified function √2sin(2x+$\frac{\pi}{4}$) has a period of π, derived from the standard sine period formula 2π/|B|. 
 3. On the interval x∈[−$\frac{\pi}{4}$, $\frac{\pi}{4}$], the argument 2x+$\frac{\pi}{4}$ ranges from −$\frac{\pi}{4}$ to $\frac{3\pi}{4}$, covering key monotonic segments of the sine function. 
 4. The maximum value of f(x) is √2 (when sin(·)=1) and the minimum is −1 (at the left endpoint where sin(−$\frac{\pi}{4}$)= $-\frac{\sqrt{2}}{2}$). |
| EBR | 15.5082 | 4367 | 0.35 | 0.66 | 1. The equation is analyzed by comparing the linear function f(x)=x/100 with the periodic sine function g(x)=sinx. 
 2. Intersections occur only when $|x|\le 100$, since the sine function is bounded between -1 and 1. 
 3. Within this range, sinxsinx completes about 31.83 periods, suggesting approximately 64 intersections (two per period plus endpoints). 
 4. Python code confirms the actual number of intersections is 63, refining the initial estimate. |

Figure 7: Strategy preference data collection showing multi-strategy performance evaluation for a trigonometric function problem. Each strategy exhibits distinct profiles across the three evaluation dimensions: correctness, process quality, and computational efficiency.

## A.3 Sensitivity Analysis of Score Aggregation Weights

The suitability score in Equation 4 aggregates three evaluation dimensions through weighted combination. To determine the optimal weight configuration, we conducted systematic ablation experiments on the same 200-problem validation set used for threshold optimization. We evaluated seven configurations representing different design choices: correctness-prioritized (0.70, 0.15, 0.15), balanced-moderate (0.60, 0.20, 0.20), fully uniform (0.33, 0.33, 0.33), quality-prioritized (0.20, 0.60, 0.20), efficiency-prioritized (0.20, 0.20, 0.60), quality-extreme (0.15, 0.70, 0.15), and

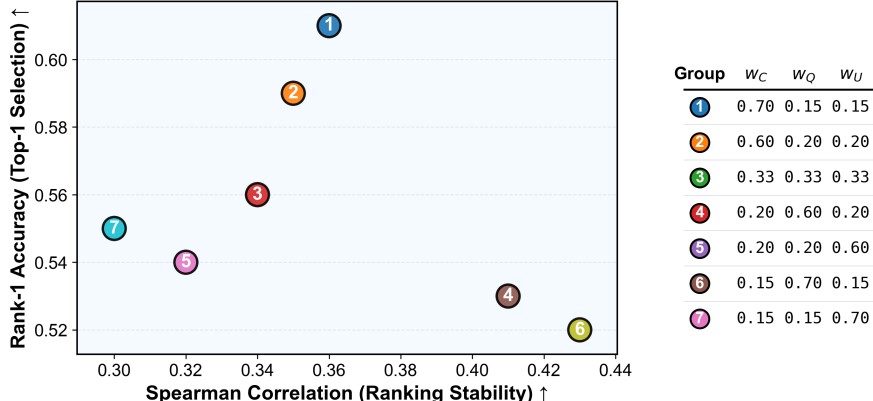

Figure 8: Sensitivity analysis of score aggregation weights on validation set. The scatter plot shows Rank-1 Accuracy versus Spearman Correlation across seven weight configurations.

efficiency-extreme (0.15, 0.15, 0.70). For each configuration, we measure Rank-1 Accuracy (percentage of problems where the top-ranked strategy matches the ground-truth best) and Spearman Correlation (rank correlation between predicted and ground-truth strategy rankings).

Figure 8 presents the results across the accuracy-stability space. The correctness-prioritized configuration (0.70, 0.15, 0.15) achieves the highest Rank-1 accuracy at 61.0%, outperforming quality-prioritized weighting (52.0%) by 9 percentage points. This substantial difference validates emphasizing correctness, which directly determines whether the selected strategy successfully solves the problem. Higher correctness weights improve Top-1 selection accuracy but reduce ranking stability. This reflects our focus on selecting the best strategy for deployment rather than achieving perfect ranking consistency.The moderate Spearman correlations (0.30-0.43) are expected given the limited ranking space with only 4 strategies and the similarity in strategy performance on individual problems. Configurations maintaining correctness weights between 0.33-0.70 demonstrate robust performance (56-61%), indicating stability within reasonable parameter ranges while exhibiting a clear optimal region. We select $w_C = 0.70$, $w_Q = 0.15$, $w_U = 0.15$ based on its empirical superiority and theoretical alignment with mathematical reasoning evaluation practices that prioritize correctness.

### A.4 THRESHOLD PARAMETER OPTIMIZATION

To determine the optimal confidence threshold $\tau_c$ and ambiguity margin $\tau_a$ for our adaptive routing policy, we conducted a comprehensive grid search on a validation set comprising 200 problems sampled from MATH, GSM8K, AQUA-RAT, SVAMP, and ASDiv datasets to ensure coverage across different mathematical domains and difficulty levels. We evaluated $\tau_c \in [0.1, 0.7]$ and $\tau_a \in [0.02, 0.20]$ with step sizes of 0.05 and 0.01 respectively. Figure 9 shows the parameter optimization results across the two-dimensional parameter space. The contour plot reveals a well-defined optimal region at $\tau_c = 0.4$ and $\tau_a = 0.08$, achieving 78.0% Pass@1 accuracy on the validation set. The performance landscape exhibits several notable characteristics:

**Confidence Threshold Sensitivity**: The $\tau_c$ parameter shows an inverted-U relationship with performance (Figure 9, top right panel). Very low confidence thresholds ($\tau_c < 0.2$) result in overconservative routing that fails to leverage high-confidence predictions effectively, achieving only 65% accuracy. Conversely, excessively high thresholds ($\tau_c > 0.5$) force the system into single-strategy execution even for uncertain predictions, degrading performance to 71%.

**Ambiguity Margin Sensitivity**: The $\tau_a$ parameter demonstrates a sharp optimum (Figure 9, bottom right panel). The optimal value of 0.08 creates an appropriate balance for distinguishing competitive strategy scenarios. Lower values ($\tau_a < 0.06$) cause excessive deliberative routing even when strategy preferences are clear, while higher values ($\tau_a > 0.12$) prevent beneficial dual-strategy verification in genuinely competitive cases.

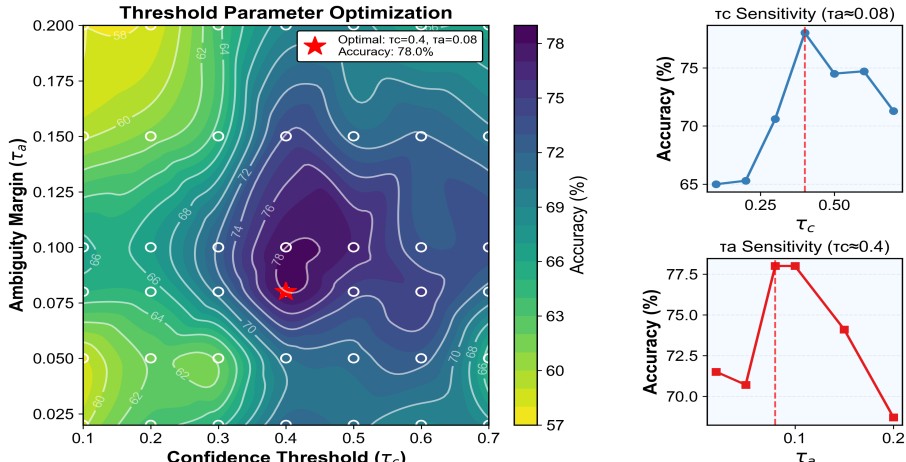

Figure 9: Threshold parameter optimization on validation set. Left: Contour plot showing accuracy across the $\tau_c$-$\tau_a$ parameter space with optimal point marked by red star. Right: Sensitivity analysis showing accuracy curves for individual parameters while holding the other at optimal value. The validation set consists of 200 problems sampled across MATH, GSM8K, AQUA-RAT, SVAMP, and ASDiv to ensure diverse difficulty coverage.

The relatively narrow optimal region indicates that careful parameter tuning is essential for achieving peak performance. However, the clear convex structure around the optimum suggests stable convergence during hyperparameter search. We use these validated parameters ($\tau_c = 0.4, \tau_a = 0.08$) across all experimental settings to ensure fair comparison with baseline methods.

### A.5 STRATEGY PERFORMANCE AND CORRELATION ANALYSIS

Figure 10 presents the mean performance scores and inter-strategy correlations across four mathematical reasoning datasets. The left panel shows that all strategies achieve similar average suitability scores (ranging from 0.18 to 0.31), with no single strategy demonstrating clear dominance across datasets. The right panel displays correlation matrices revealing predominantly low or negative correlations between strategies, indicating complementary rather than redundant capabilities. Notably, TIR exhibits consistent negative correlations with other strategies across most datasets, suggesting its specialized applicability to distinct problem characteristics. These patterns validate the necessity of adaptive strategy selection, as the low inter-strategy correlations demonstrate that different approaches excel on different problem subsets.

### A.6 CASE STUDY: STRATEGY ADAPTER PREDICTION

Figure 11 presents a representative example demonstrating how our Strategy Adapter evaluates different reasoning approaches for a logarithmic geometry problem. The problem requires finding the x-coordinate where a horizontal line intersects the curve $f(x) = \ln x$, involving both coordinate geometry concepts and logarithmic calculations.

The Strategy Adapter assigns suitability scores that align well with actual strategy performance: TIR receives the highest score (0.63) and successfully solves the problem through tool-assisted computation, while NLR and CAR receive lower scores (0.28 and 0.16 respectively) and both fail to produce correct solutions. EBR achieves a moderate score (0.58) and succeeds through ensemble reasoning. This case exemplifies how the adapter learns to associate problem characteristics—such as the need for precise numerical computation in logarithmic contexts—with appropriate reasoning strategies.

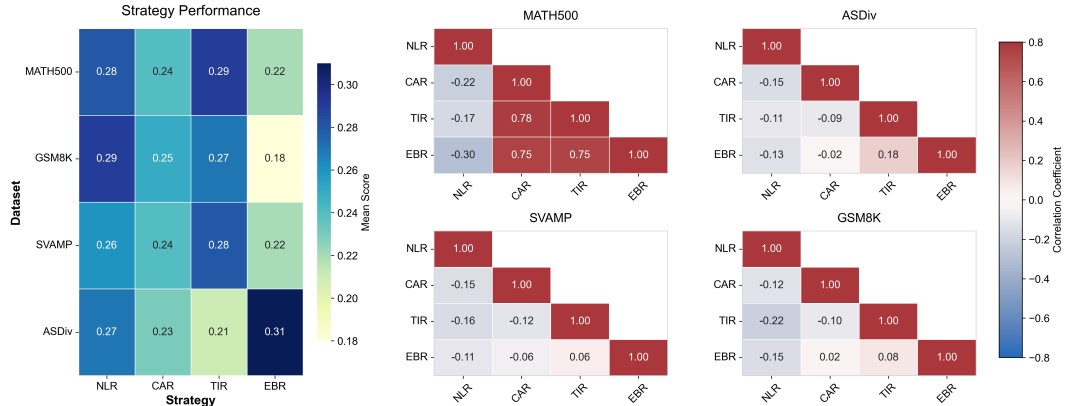

Figure 10: Strategy performance and correlation analysis. Left: Mean suitability scores by strategy and dataset. Right: Inter-strategy correlation matrices for each dataset. Low correlations indicate complementary strategy capabilities.

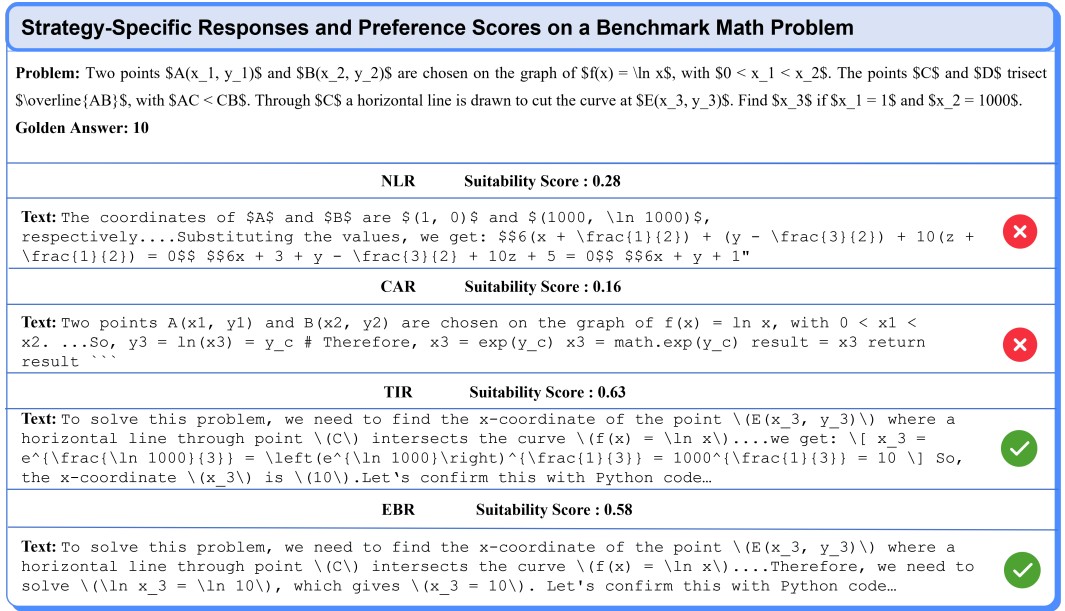

Figure 11: Case study showing Strategy Adapter suitability scores and corresponding strategy outcomes for a logarithmic geometry problem. Higher suitability scores correlate with successful problem-solving performance.

The correlation between prediction scores and actual outcomes validates our approach of using suitability scores to guide adaptive routing decisions, demonstrating that higher Strategy Adapter scores generally correspond to better strategy performance on specific problem instances.

## A.7 ERROR CASE ANALYSIS

To better understand PRISM's limitations, we analyze representative failure cases that reveal two primary error patterns. Figure 12 illustrates a high-confidence routing error where the Strategy Adapter assigns the highest score to TIR (0.357) for a trigonometric product calculation problem. Despite this confident prediction, TIR produces an incorrect numerical result (0.0195 instead of $\frac{9}{256}$), demonstrating that high discriminator scores do not always guarantee correct solutions. This case reveals a calibration issue where the adapter overestimates strategy suitability based on problem

**Error Case Analysis – High-Confidence Routing Errors**

**Problem:** Find $\sin 20^\circ \sin 40^\circ \sin 60^\circ \sin 80^\circ \sin 100^\circ \sin 120^\circ \sin 140^\circ \sin 160^\circ.$

**Golden Answer:** \frac{9}{256}

**Mode:** prism

**Chosen_path:** prism-single-path

**Strategies_actually_executed_in_path:** [TIR]

**Discriminator_scores:** { NLR: 0.236978, CAR: 0.221464, TIR: 0.357294, EBR: 0.184264 }

**Raw_llm_output :** \nTo solve the problem, we can use trigonometric identities and properties. The given expression is:\n\n\\[\n\\sin 20^\\circ \\sin 40^\\circ \\sin 60^\\circ \\sin 80^\\circ \\sin 100^\\circ \\sin 120^\\circ \\sin 140^\\circ \\sin 160^\\circ\n\\]\n\nWe can use the identity …,Now we need to find \\((\\sin 20^\\circ)^2 (\\sin 40^\\circ)^2\\). We can use the product-to-sum identities. Specifically, …,Now we can multiply this by \\(\\frac{3}{4}\\):\n\n\\[\n(\\sin 20^\\circ)^2 (\\sin 40^\\circ)^2 (\\sin 60^\\circ)^2 = \\left(\\frac{1}{2} \\cos(20^\\circ) - \\frac{1}{4}\\right)^2 \\cdot \\frac{3}{4}\n\\]\n\nThis expression is quite complex to simplify further without numerical evaluation. However, we can use Python to evaluate this expression accurately. Let's use Python to compute the value.\n```python\nimport math\n\n# Define the angles in degrees\nangles = [20, 40, 60, 80, 100, 120, 140, 160]\n\n# Convert degrees to radians\nangles_rad = [math.radians(angle) for angle in angles]\n\n# Calculate the product of sines,…,\n\nprint(product_of_sines)

**Extracted_answer: 0.01953125**

Figure 12: High-confidence routing error: Strategy Adapter assigns highest score to TIR, which executes but produces incorrect numerical result.

**Error Case Analysis – Aggregation Errors**

**Problem:** Let $z = 2 + \sqrt{2} - (3 + 3 \sqrt{2})i$, and let $c = 2 - 3i$. Let $w$ be the result when $z$ is rotated around $c$ by $\frac{\pi}{4}$ counter-clockwise.\n\n[asy]\n unitsize(0.6 cm);\n\n pair C, W, Z;\n\nZ = (2 + sqrt(2), -3 - 3*sqrt(2));\nC = (2,-3);\nW = rotate(45,C)*(Z);\n\ndraw(Z--C--W);\n\n dot(\"$c$\", C, N);\n dot(\"$w$\", W, SE);\ndot(\"$z$\", Z, S);\nlabel(\"$\\frac{\pi}{4}$\", C + (0.6,-1));\n[/asy]\n\n Find $w.$

**Golden Answer: 6 - 5i**

**Mode:** prism

**Chosen_path:** prism-dual-path

**Strategies_actually_executed_in_path:** [CAR, NLR]

**Discriminator_scores:** { NLR: 0.284763, CAR: 0.329766, TIR: 0.20969, EBR: 0.175781 }

**Strategy_name:** CAR
**Raw_llm_output:** import cmath\n\n # Given complex numbers\n z = 2 + cmath.sqrt(2) - (3 + 3 * cmath.sqrt(2)) * 1j\n  c = 2 - 3* 1j\n\n # Rotate z around c by pi/4 counter-clockwise\n  angle = cmath.pi / 4\n  w = c + (z - c) * cmath.exp(1j * angle)\n\n result = w.real, w.imag\n  return result\n```\n
**Extracted_answer: (6,-5.000000000000001)**
**Is_correct_intermediate_answer: FALSE**

**Strategy_name:** NLR
**Raw_llm_output:** Let's think step by step\nWe write $z$ as $2 + \\sqrt{2} - (3 + 3 \\sqrt{2})i$ and $c$ as $2 - 3i$.\nThe difference $z - c$ is …,To rotate this complex number by $\\frac{\pi}{4}$, we multiply by …,This gives…,This gives…, we add $c$ to get $w = (4 - 2i) + (2 - 3i) = \\boxed{6 - 5i}$.\nThe answer is 6-5i\n
**Extracted_answer: 6-5i**
**Is_correct_intermediate_answer: TRUE**

Figure 13: Aggregation error: In dual-path routing, CAR (higher score) produces malformed output while NLR (lower score) generates correct answer, but confidence-based selection chooses incorrectly.

surface features (presence of trigonometric functions suggesting tool usage) without anticipating execution failures in symbolic manipulation.

Figure 13 presents an aggregation error in dual-path routing. For a complex number rotation problem, the adapter correctly identifies CAR (0.330) and NLR (0.285) as competitive strategies. However, while both strategies are executed, CAR produces a formatting error (6,-5.0000000000000001) and NLR generates the correct answer (6-5i). The routing policy selects CAR's output based on higher predicted confidence, resulting in failure. This case highlights a limitation in our confidence-based aggregation mechanism: when strategies produce inconsistent answers, prediction scores alone may not reliably indicate correctness. These failure patterns suggest that future work could benefit from incorporating runtime validation signals or more sophisticated answer consistency checking beyond confidence-weighted selection.

