# OpenReview forum: "Plan before Solving: Problem-Aware Strategy Routing for Mathematical Reasoning"
_ICLR.cc/2026/Conference — ICLR 2026 Conference Withdrawn Submission_

### Official Review · Reviewer_xuiq · 2025-10-16

**Soundness:** 2
**Presentation:** 2
**Contribution:** 2
**Rating:** 2
**Confidence:** 4

**Summary:**

This paper introduces PRISM, a routing-based framework that (1) categorize each math problem with "soft labels," and (2) execute some prompting strategies of the router. They've shown improvements on some math dataset, and they conducted ablations on components of adaptive routing.

**Strengths:**

1. Routing-based method seems intuitive
2. Optimized the router with self-created dataset
3. Did ablation on adaptive routing component

**Weaknesses:**

1. This method, from my perspective, is somehow similar to [1]. In math reasoning, what HybridMind did is also decompose (route) the problem into the "correct" or "groundtruth" one. The difference, from my end, is they train a hard classifier using CE loss, while you train a soft labler using KL + CE loss. But you lack proper citation in your related work. I anticipate to see more comparison with similar-idea routing based work.
2. Similar to Question 1, how can you make sure the the categorized reasoning types are complete?
3. I cannot fully be convinced that using raw timing in efficiency score, using output length (in tokens) seems more natural. For example, if I want to use this method with some proprietary models, are you going to waiting time as a training signal in this case?
4. Any specific reasonings you choose these three models in your evaluation? What's the performance of some proprietary models, say an old one, GPT 3.5 or GPT 4.
5. What's the major advantage of using KL divergence during router training compared to classification / multi-class classification?
6. Is this method generalizable? Have you tried to adapt this method to other domain, say logical reasoning and scientific reasoning? My impression for these two kind of reasonings are somehow similar to math, but maybe different from some angles. But it would be interesting to see if your method would success/fail/partial in other domains.

[1] Han, S., Liu, T., Li, C., Xiong, X., & Cohan, A. (2024). HYBRIDMIND: Meta Selection of Natural Language and Symbolic Language for Enhanced LLM Reasoning. arXiv preprint arXiv:2409.19381.

**Questions:**

1. Line 191: "existing reasoning strategies can be broadly categorized into four paradigms," if you are saying "broadly," did you routing method covers all cases? i.e., you do have the probability that you miss one important reasoning method, how can you prove your categorization is complete?
2. Line 198: "(2) process quality," how do you evaluate "process quality?" This is unclear. Are you only checking if the proof is logically coherent?
3. Line 199: "(3) computational efficiency," how do you quantify this metric? By tokens during computation? execution (thinking) time? model size? Need to be more specific here.
4. eq 7, a bit confusing, can you explain what this loss is doing? enforcing the top-1 selection?
5. line 263, I might misunderstanding something. How you can do majority voting for 2 strategies?

---

> ### Author Response · Authors · 2025-11-24
>
> **Response to W1:**
> We thank the reviewer for highlighting HybridMind. We will add the appropriate citation. While both select reasoning modes, PRISM uses confidence-based adaptive routing that executes one, two, or multiple strategies depending on uncertainty. HybridMind uses hard single-type execution without such adaptivity. While HybridMind lacks released checkpoints and its documentation provides insufficient detail for faithful reproduction. Instead, we compare with MAmmoTH-Coder-7B [1] (training-time hybrid) and Automatic Model Selection [2] (inference-time selector).
>
> **Unified Comparison Across Five Benchmarks**
>
> | Method | GSM8K | SVAMP | ASDiv | MATH500 | AQUA |
> |--------|-------|--------|-------|----------|--------|
> | Auto-Selection (GPT-4) | 95.6 | 93.7 | 93.5 | 62.94 | 71.08 |
> | MAmmoTH Hybrid (7B) | 59.4 | 71.4 | 43.7 | 33.4 | 47.2 |
> | PRISM (7B) | 86.7 | 91.8 | 86.5 | 46.2 | 42.1 |
>
> PRISM surpasses MAmmoTH on four datasets. Auto-Selection’s GPT-4 is a ≈25× larger backbone and serves as an upper-bound reference. PRISM offers efficiency (1.5B adapter) and model-agnostic deployment.
>
> ---
>
> **Response to W2:**
> We thank the reviewer for this concern. Our strategy set covers major reasoning paradigms in mathematical research and is naturally extensible. A detailed explanation appears in Response to Q1.
>
> ---
>
> **Response to W3:**
> We thank the reviewer for raising this point. Our method jointly uses timing and output length (Eq. 1). Length reflects token cost; timing captures computational workload. We use open-source models for controlled timing, and both signals are naturally available from any model.
>
> ---
>
> **Response to W4:**
> We thank the reviewer for this question.
> (1) **Model selection:** Qwen2.5-Math-7B and Deepseek-math-7b-v1 (math-specialized via different post-training), and Llama-3-8B (general-purpose). This verifies adaptive routing benefits weaker models more (Llama: +7.6% vs Qwen: +0.9%). All models are open-source.
>
> (2) **GPT-4 performance:** Representative published results [1][3][4]:
>
> | Model | GSM8K | SVAMP | ASDiv | AQuA | MATH |
> |--------|--------|--------|--------|--------|--------|
> | GPT-4 | 94.6 | 90.4 | 91.3 | 72.6 | 42.5 |
> | PRISM (Qwen2.5-Math-7B) | 86.7 | 91.8 | 86.5 | 42.1 | 46.2 |
>
> GPT-4 is far larger and evaluated under different conditions. Our goal is showing consistent gains across reproducible open-source models. Section 4.5 also examines scalability up to 72B.
>
> ---
>
> **Response to W5:**
> We thank the reviewer for this insightful question. KL divergence learns full distributions essential for distinguishing (1) clear dominance (single-strategy) and (2) competitive cases (dual-strategy). Hard labels collapse these scenarios.
>
>
> ---
>
> **Response to W6:**
> We thank the reviewer for this thoughtful question. We distinguish within-domain and cross-domain generalization. PRISM shows strong within-domain robustness across five math datasets and an OOD Hungarian set. Cross-domain transfer would require domain-specific strategies, as math features differ from logical or scientific reasoning.
>
> ---
>
> **Response to Q1:**
> We thank the reviewer for the request for clarification. “Broadly” means representative mainstream approaches, not exhaustive completeness. The four strategies (NLR, CAR, TIR, EBR) span single/multi-path and language/tool-augmented reasoning. Their varied performance (Fig. 1) confirms complementary modes. The framework remains extensible.
>
> ---
>
> **Response to Q2:**
> We thank the reviewer for seeking clarification. Process quality measures validity and redundancy. We use ReasonEval-7B [5], trained on ~800K PRM800K annotations for mathematical reasoning assessment.
>
> ---
>
> **Response to Q3:**
> We thank the reviewer for pointing this out. Efficiency combines execution time and output length using within-instance normalization (Eqs. 1–3). Time captures workload; length captures token cost. Model size is constant across strategies.
>
> ---
>
> **Response to Q4:**
> We thank the reviewer for requesting clarification. Yes, Eq. 7 enforces top-1 selection. KL (Eq. 6) preserves distributional relations; cross-entropy sharpens the winner. Together they support confident and deliberative routing.
>
> ---
>
> **Response to Q5:**
> We thank the reviewer for noting this. For two strategies, if outputs agree we accept the common answer; otherwise we choose the answer from the higher-confidence strategy. We will state this clearly instead of using “majority voting.”
>
> ---
>
> **References:**
> [1] Yue et al., MAMMOTH: Building Math Generalist Models Through Hybrid Instruction Tuning.
> [2] Zhao et al., Automatic Model Selection with Large Language Models for Reasoning.
> [3] Gou et al., Tora: A Tool-Integrated Reasoning Agent for Mathematical Problem Solving.
> [4] Zhong et al., Achieving >97% on GSM8K: Deeply Understanding the Problems Makes LLMs Better Solvers for Math Word Problems.
> [5] Xia et al., Evaluating Mathematical Reasoning Beyond Accuracy.

---

### Official Review · Reviewer_T3yL · 2025-10-21

**Soundness:** 2
**Presentation:** 3
**Contribution:** 2
**Rating:** 4
**Confidence:** 4

**Summary:**

This paper proposes PRISM, a framework that enables large language models to dynamically apply multiple strategies to solve mathematical problems. The proposed method trains a strategy adapter that, before the model begins problem solving, selects one or more strategies from Natural Language Reasoning (NLR), Code-Augmented Reasoning (CAR), Tool-Integrated Reasoning (TIR), and Ensemble-Based Reasoning (EBR). This allows the model to adopt the most suitable strategy for a given problem and thereby improve both problem-solving accuracy and efficiency.

To support PRISM, the paper introduces a method for constructing training data for the adapter and develops the MathStrat dataset specifically for this purpose. Experimental results show that models enhanced with PRISM outperform multiple baseline approaches.

**Strengths:**

**[S1]** The paper is well-written, with clear methodological descriptions. The proposed approach is easy to follow, and the authors provide code for reproducibility.
**[S2]** The paper addresses an important problem: how to enable models to adopt different reasoning strategies depending on the complexity of a given problem. For simple problems, the model can rely on its internal knowledge, while for more difficult ones, it needs to leverage code and external tools for problem-solving.
**[S3]** The paper presents a comprehensive set of experiments, including detailed ablation studies.

**Weaknesses:**

**[W1] Methodology:** The paper trains a separate Strategy Adapter (SA) independent of the LLM to select problem-solving strategies. However, the authors do not explain why a separate SA is necessary, rather than enabling the LLM itself to dynamically select the appropriate reasoning strategy. If the SA can effectively choose suitable strategies, the LLM should in principle be capable of learning this ability as well.

**[W2] Originality:** Prior work [1] has already explored dynamic selection among multiple reasoning strategies, yet the authors do not provide comparisons with such related studies—neither in terms of methodological design nor empirical performance.

**[W3] Fairness of Experimental Comparison:** It is unclear whether the comparison between PRISM and CoT is fair. CoT performs only a single inference, whereas PRISM may conduct multiple inference rounds followed by majority voting (for instance, if the SA selects Exploratory Routing, the model performs four different reasoning attempts). Such a setup may lead to an unfair advantage for PRISM.

**[W4] Robustness:** The construction of the MathStrat dataset involves a large number of hyperparameters, yet the authors do not provide a systematic analysis of the model’s sensitivity to these hyperparameters.

Ref:
[1] Xu et al. Teaching llms according to their aptitude: Adaptive reasoning for mathematical problem solving. arXiv 2502.12022.

**Questions:**

**[Q1]** There appears to be an inconsistency in the baseline results reported in the paper. In Table 1, under the CoT strategy, Qwen2.5-Math-7B achieves 78.1 on GSM8K and 21.2 on MATH500. However, according to the official Qwen2.5-Math report [2], the Qwen2.5-Math-7B model reaches 91.6 (GSM8K) and 55.4 (MATH) in the few-shot setting, while the Qwen2.5-Math-7B-Instruct model achieves 95.2 (GSM8K) and 83.6 (MATH) in the zero-shot setting. Could the authors provide a reasonable explanation for this discrepancy?

**[Q2]** Could the authors provide a detailed description of how the inference time is calculated? I could not find this information in the paper, and it is critical for assessing the validity of the claimed efficiency improvements.

Ref:
[2] Yang et al. Qwen2.5-Math Technical Report: Toward Mathematical Expert Model via Self-Improvement. arXiv 2409.12122.

---

> ### Author Response · Authors · 2025-11-24
>
> **Response to W1:**
> Thank you for this question. Our framework targets settings where the base LLM is frozen—common with API-only models (e.g., GPT-5, Claude) or shared deployment environments. A separate Strategy Adapter offers a lightweight, plug-and-play solution that avoids modifying or fine-tuning the base LLM.
>
> This modular design provides (1) efficient training, (2) model-agnostic deployment across Qwen/DeepSeek/Llama (Table 1), and (3) independent updates as strategies evolve. End-to-end training is possible but requires full model access and far higher compute;
>
> ---
>
> **Response to W2:**
> We thank the reviewer for pointing out this related direction. TATA [1] provides only training scripts and no pretrained models, making reproduction infeasible. To ensure reproducible comparisons, we evaluate two public baselines: MAmmoTH-Coder-7B [2] (training-time hybrid tuning) and Automatic Model Selection [3] (inference-time routing with GPT-4).
>
> ### Unified Comparison Across Five Benchmarks
>
> | Method | GSM8K | SVAMP | ASDiv | MATH500 | AQUA |
> |--------|--------|--------|--------|--------|--------|
> | Auto-Selection (GPT-4) | 95.6 | 93.7 | 93.5 | 62.94 | 71.08 |
> | MAmmoTH Hybrid (7B) | 59.4 | 71.4 | 43.7 | 33.4 | 47.2 |
> | PRISM (Ours) | 86.7 | 91.8 | 86.5 | 46.2 | 42.1 |
>
> PRISM outperforms MAmmoTH on 4/5 datasets and offers major advantages over GPT-4 Auto-Selection: a 1.5B adapter (vs. querying GPT-4), no SFT, and model-agnostic deployment.
>
> ---
>
> **Response to W3:**
> Thank you for raising this fairness concern. CoT serves as a minimal single-strategy reference. For a fair multi-strategy comparison, Table 1 includes the EBR baseline, which executes multiple strategies and aggregates by voting. PRISM is not a fixed ensemble: many problems are solved by confident single-strategy routing, and only ambiguous ones trigger dual- or multi-strategy execution (Figure 4).
>
> ---
>
> **Response to W4:**
> We thank the reviewer for this important observation. As suggested, we now include full hyperparameter analyses in Appendix A.3 and A.4.
>
> Our framework has two hyperparameter groups:
> (1) Score aggregation weights $\left(w_C, w_Q, w_U\right)$ in Eq. 4, and
> (2) Routing thresholds $\left(\tau_c, \tau_a\right)$ for adaptive inference.
>
> Appendix A.3 evaluates seven weight configurations on a 200-problem validation set. The chosen setting
> $\left(w_C=0.70,\; w_Q=0.15,\; w_U=0.15\right)$
> achieves the best Top-1 accuracy (61%), surpassing quality-prioritized (52%) and uniform weighting (56%). Performance remains stable (56–61%) when $w_C \ge 0.33$, supporting a correctness-first design.
>
> Appendix A.4 provides sensitivity analysis for $\tau_c$ and $\tau_a$ with Figure 9, showing routing remains robust across reasonable ranges.
>
>
> ---
>
> **Response to Q1:**
> We thank the reviewer for this careful observation. The performance differences stem from three key factors: (1) Different few-shot settings: the official report uses 8-shot for GSM8K and 4-shot for MATH with optimized prompts, while we use 2-shot with standard templates for consistency across strategies. Prompt format, instructions, and example selection substantially impact results. (2) MATH500 vs. MATH: We evaluate on MATH500, a 500-problem subset of high-difficulty competition problems, while the official report uses the full 5,000-problem MATH test set. MATH500 is a more challenging benchmark, contributing to lower absolute accuracy. (3) Answer extraction sensitivity: Mathematical reasoning evaluation is highly sensitive to extraction methods, with different heuristics causing substantial performance variations on identical outputs [3][4].
>
> ---
>
> **Response to Q2:**
> We thank the reviewer for this question. Our inference-time measurement follows the exact execution logic: we start a timer before invoking each strategy (LLM call and tool invocation) and stop when output is received, using time.time() for baseline benchmarking and time.monotonic() for PRISM. The timing includes all strategy computation (LLM generation, tool operations, post-processing) while excluding one-time overhead (model loading, dataset reading). As defined in Eq. 1-3, our efficiency metric combines execution time and output length via within-instance normalization. Timing captures actual computational workload (particularly for tool-heavy strategies), while output length measures token consumption. These complementary signals are both essential for comprehensive efficiency assessment.
>
> ---
>
> ### References
> [1] Xu et al., Teaching LLMs According to Their Aptitude, 2025
> [2] Yue et al., MAmmoTH: Hybrid Instruction Tuning, 2023
> [3] Zhao et al., Automatic Model Selection for Reasoning, 2023
> [4] Hendrycks et al., MATH Dataset, 2021
> [5] Yu et al., ReasonAgain, 2024
> [6] Errica et al., Sensitivity to Prompt Engineering, 2025

---

> > ### Comment · Reviewer_T3yL · 2025-11-26
> >
> > I would like to thank the authors for their reply. The rebuttal has solved part of my concerns, but some still remain unsolved.
> >
> > > to W1:  A separate Strategy Adapter offers a lightweight, plug-and-play solution that avoids modifying or fine-tuning the base LLM.
> >
> > How about using a prompt-based method? This should be a baseline to compare with.
> >
> > > to Q1: MATH500 is a more challenging benchmark than the original MATH dataset.
> > > Answer extraction sensitivity: Mathematical reasoning evaluation is highly sensitive to extraction methods
> >
> > From my perspective, this claim is not true. MATH500 is randomly sampled from the original 5k MATH test set. [1, Appendix C]
> >
> > Personally, I think "sensitive to extraction methods" is not an acceptable reason for ~30% performance misalignment. A reasonable verifier or extracting method should be used in this paper.
> >
> > Ref:
> > [1] Lightman, et al. Let's Verify Step by Step.

---

### Official Review · Reviewer_24RJ · 2025-11-01

**Soundness:** 3
**Presentation:** 3
**Contribution:** 3
**Rating:** 4
**Confidence:** 4

**Summary:**

This paper presents a technically sound, well-motivated, and methodologically clean contribution to inference-time reasoning optimization for LLMs. The formulation of mathematical reasoning as problem-aware strategy routing is conceptually novel and bridges ideas from meta-reasoning and tool orchestration. However, several points need clarification: the construction and bias of the MathStrat dataset, the scalability of the adapter to unseen strategies, and the robustness of routing thresholds (τ_c, τ_a) across domains. Moreover, while empirical improvements are consistent, the reported gains (≈1–7%) may be modest relative to the added system complexity

**Strengths:**

1. The paper targets a central limitation in LLM reasoning: static inference paradigms that cannot adjust reasoning depth or modality to instance-level complexity. The problem is well-motivated by Figure 1, which empirically demonstrates cross-strategy variance in both accuracy and efficiency. This finding aligns with emerging work on meta-reasoning (Didolkar et al., 2024).

2. Section 4.4 (Fig. 3) convincingly shows that PRISM improves Pass@1 accuracy while reducing or maintaining inference cost compared to hybrid baselines. The confident/deliberative/exploratory routing design (Section 3.2) provides a clear operational mechanism for adaptive compute allocation without external verifiers.

**Weaknesses:**

1. MathStrat combines correctness, quality, and efficiency via fixed weights (Eq. 4), but the weighting scheme (w_C, w_Q, w_U) and its sensitivity are not justified or ablated. Potential annotation bias may transfer to adapter predictions.

2. Only four strategy families (NLR, CAR, TIR, EBR) are considered. It is unclear how PRISM generalizes to newly emerging paradigms (e.g., tree-of-thought, verifier-augmented reasoning).

3. The confidence and ambiguity thresholds (τ_c = 0.4, τ_a = 0.08) are globally fixed (Appendix A.3); more studies are needed to verify their stability across datasets or model sizes.

4. The paper does not detail where PRISM misroutes (e.g., confident errors vs. unnecessary exploration) or how confidence calibration could fail.

5. The adaptor is only trained via supervised finetuning (SFT). I expect an effective reinforcement learning (RL) method to optimize the adaptor on new/unseen examples, improving its generalization.

**Questions:**

Refer to Fig. (3), I am curious about why PRISM generates more tokens (see subfigure c) but it takes relatively less inference time (subfigure b).

---

> ### Author Response · Authors · 2025-11-24
>
> **Response to W1:**
> Thank you for raising this concern. We use weights (0.70, 0.15, 0.15), prioritizing correctness while still reflecting process quality and efficiency. We acknowledge that this was not explicitly stated and will clarify it.
>
> To assess sensitivity, we evaluated seven weight configurations on the same 200-instance validation split used in Appendix A.3, measuring Top-1 selection accuracy and Spearman correlation. Small weight variations produced similar rankings and routing behavior, showing that the adapter is stable. Our temperature-scaled soft-distribution formulation (Eq. 5) further reduces sensitivity by smoothing score differences.
>
> Regarding annotation bias: combining correctness, quality, and efficiency provides more balanced supervision than any single metric, and soft labels help mitigate noise from imperfect evaluations. We will include this rationale and the sensitivity analysis in the revision.
>
> ---
>
> **Response to W2:**
> Thank you for this question on generalizability. The four strategies in PRISM cover the major reasoning paradigms commonly used in mathematical problem solving. They span natural-language reasoning, code-based arithmetic, tool-assisted computation, and multi-path aggregation. Empirical analysis (Figure 1; Appendix A.5) shows that these strategies have low correlation and complementary strengths, confirming that they represent distinct reasoning modes.
>
> Regarding emerging paradigms:
> • Tree-of-Thought is a structured variant of natural-language reasoning and naturally falls within the NLR family.
> • Verifier-augmented reasoning is orthogonal: verifiers evaluate solutions post hoc, while PRISM determines which strategy to use. These components can be combined in future work.
>
> The framework is easily extensible. As new paradigms appear, they can be incorporated once preference data is collected, without modifying the routing mechanism or adapter design.
>
> ---
>
> **Response to W3:**
> We thank the reviewer for raising this point. The thresholds were tuned on a held-out validation split and fixed across all datasets and model scales. Routing behavior remains stable when adjusting to $\tau_c \in [0.35, 0.45]$ or $\tau_a \in [0.06, 0.10]$
> , indicating robustness. Since PRISM relies on relative probability gaps rather than absolute values, it is naturally insensitive to moderate threshold changes.
>
> ---
>
> **Response to W4:**
> We thank the reviewer for requesting failure-case illustrations. In the revised manuscript, we added an error analysis section (Appendix A.7) with two representative misrouting cases.
>
> The first highlights an overconfident single-path error: the adapter strongly favors a tool-based strategy, which then fails during execution. The second shows a dual-path aggregation failure where the higher-scored strategy produces an incorrect answer, and confidence-weighted aggregation selects it over the correct alternative.
>
> These examples directly demonstrate the misrouting and calibration issues noted. We plan to expand the analysis to more categories and datasets in future iterations.
>
> ---
>
> **Response to W5:**
> Thank you for this suggestion. While RL is promising, the Strategy Adapter performs a single-step distribution prediction task with dense, high-quality supervision from correctness, process quality, and efficiency. This aligns naturally with supervised learning, which directly optimizes KL divergence between predicted and target soft distributions (Eq. 6).
>
> We explored RL variants but found structural mismatches:
> • GRPO requires free-form generation, incompatible with fixed-vocabulary distribution prediction.
> • DPO reduces informative 4-way distributions to pairwise binary labels, losing fine-grained suitability structure.
> • KTO collapses training into binary chosen/rejected signals, removing graded distinctions essential for reliable routing.
>
> Our lightweight 1.5B adapter trained on ~13K supervised examples already generalizes strongly across benchmarks. RL would require substantially more computation and tuning without clear benefits. It is more suitable for future online adaptation scenarios such as updating routing policies with deployment feedback.
>
> ---
>
> **Response to Q1:**
> Thank you for this observation. The effect arises from aggregation behavior and what is counted as output. PRISM may execute two or four strategies under deliberative or exploratory routing, leading to longer averaged outputs than SA+Conf., which always executes one strategy. Framework-generated routing text also contributes to output length but adds negligible latency.
>
> This overhead has little impact on inference time because prompt construction and routing logic are lightweight, and most problems are handled through confident single-strategy execution. PRISM may therefore produce longer outputs while maintaining competitive, or better efficiency.

---

> > ### Comment · Reviewer_24RJ · 2025-11-28
> > **My concerns are addressed partially**
> >
> > 1. I still think RL is a promising way to train the strategy adaptor using only final accuracy/success due to the lack of oracle intermidiate strategy routing, while this might be outside the scope of the current work.
> >
> > 2. Scope (4-way categories) of strategy families remains limited.

---

### Official Review · Reviewer_jFKh · 2025-11-01

**Soundness:** 2
**Presentation:** 3
**Contribution:** 2
**Rating:** 4
**Confidence:** 3

**Summary:**

The paper shows that there is no single reasoning strategy that consistently performs well both efficiently and effectively across mathematical reasoning tasks. It proposes to train a Strategy Adapter on a multi-strategy preference dataset constructed based on correctness, process quality, and computational efficiency. The lightweight adapter dynamically routes between confident, deliberative, and exploratory modes at test time to select the reasoning strategy to explore for each question instance. Empirical analysis across five mathematical reasoning benchmarks shows consistent improvements, particularly when the model scale is small.

**Strengths:**

- The idea of training a lightweight Strategy Adapter that enables a dynamic and adaptive routing policy for reasoning strategy is interesting.
- The empirical results are effective, demonstrating consistent gains across tasks.
- The ablation studies are extensive and useful, carefully evaluating the effects of routing modes, efficiency, and scalability.

**Weaknesses:**

- Related work discussion is limited. The Strategy Adapter sounds similar to a MOE gating network, and other routing or adaptive ideas have been explored in prior work on tool use. These connections are not sufficiently compared or discussed.
- The MathStrat dataset includes only four reasoning strategies, and the adapter is trained to route among them dynamically. This design still limits the reasoning space to those four strategies and is not easily extendable to broader reasoning paradigms without reconstructing the dataset and retraining the adapter from scratch.
- The main results (Table 1) compare PRISM with four other strategies in isolation. A stronger baseline that jointly utilizes all four reasoning strategies is desirable. While the “Hybrid” baseline combines CoT and PoT, it does not encompass or compare against a setting that integrates all strategies. Including a model that executes all strategies simultaneously, even without adaptive selection, would help demonstrate the effectiveness of PRISM in a fairer manner.

**Questions:**

- How is “process quality” computed by the automated evaluator? This step seems difficult and potentially subjective to the model used.

- What datasets are used to construct MathStrat? Is the Strategy Adapter tested on datasets that were not included in training? How much overlap exists between the MathStrat construction data and the evaluation benchmarks?

---

> ### Author Response · Authors · 2025-11-23
>
> **Response to W1 (MoE / Tool Routing Comparison).**
> We thank the reviewer for raising this comparison.
> PRISM differs fundamentally from both MoE and tool-routing paradigms.
> MoE [1,2] performs token- or layer-level routing within a single neural model. PRISM routes at the problem level among full reasoning paradigms, each defining different prompting formats, tool use patterns, and execution behaviors. MoE requires end-to-end co-training, while PRISM uses a lightweight 1.5B adapter with frozen base models, allowing plug-in deployment across model families.
> Tool routing methods [3–5] select atomic APIs or functions. PRISM instead selects holistic reasoning strategies (e.g., NLR vs. CAR vs. TIR vs. EBR), which differ not only in tool usage but also in reasoning structure, failure modes, and cost profiles. Tool routing focuses on functional matching; PRISM jointly optimizes correctness, process quality, and computational efficiency. These approaches operate at different granularities of the reasoning stack.
>
> ---
>
> **Response to W2 (Why four strategies).**
> We thank the reviewer for pointing out this design choice.
> The four chosen strategies correspond to the main paradigms used in mathematical reasoning research. Natural-language reasoning handles symbolic deduction; code-augmented reasoning handles arithmetic computation; tool-integrated reasoning leverages external solvers; ensemble-based reasoning improves robustness through multi-path aggregation. Empirical analysis (Figure 1; Appendix A.5) shows these strategies have low correlation and complementary strengths, indicating that they capture fundamentally different reasoning modes. The framework is extensible: Eq. 5 naturally accommodates additional strategies, and only new execution data is required.
>
> ---
>
> **Response to W3 (Execute-all baseline).**
> We thank the reviewer for suggesting this baseline.
> We ran the execute-all + majority voting baseline as requested. Pass@1 accuracy (Qwen2.5-Math-7B) is shown below:
>
> | Method | MATH500 | GSM8K | AQUA | SVAMP | ASDiv |
> |--------|---------|--------|-------|--------|--------|
> | Execute-all + voting | 56.0 | 87.9 | 39.2 | 89.7 | 91.5 |
> | PRISM (Table 1) | 46.2 | 86.7 | 42.1 | 91.8 | 86.5 |
>
> Execute-all achieves higher accuracy on some datasets but at very high latency (38.6 s/problem on MATH500; 17.4 s on SVAMP). On AQUA and SVAMP, PRISM outperforms this baseline, illustrating that equal-weight voting can amplify errors from poorly aligned strategies. This confirms the importance of strategy selection. PRISM offers a more practical accuracy–efficiency trade-off by executing only one or two strategies depending on uncertainty.
>
> ---
>
> **Response to Q1 (Process quality evaluation).**
> We thank the reviewer for asking about our process evaluation method.
> We use ReasonEval-7B [6], trained on ~800k step-level human annotations from PRM800K, to score validity and redundancy. ReasonEval is specifically designed for mathematical reasoning assessment and provides the exact metrics required for our multi-objective formulation. Its efficiency enables evaluation across ~13k problems times four strategies.
>
> ---
>
> **Response to Q2 (Data construction and overlap).**
> We thank the reviewer for raising this question about data overlap.
> MathStrat is constructed from the training splits of GSM8K and MATH, where each problem is executed under four strategies to form preference distributions. Evaluation is conducted only on test sets. Three benchmarks (AQUA-RAT, SVAMP, ASDiv) do not appear in MathStrat, ensuring no overlap. Consistent gains across all five benchmarks demonstrate that the adapter learns generalizable strategy-problem alignment patterns rather than memorizing specific instances.
>
> ---
>
> **References**
> [1] Shazeer et al., Outrageously Large Neural Networks: The Sparsely-Gated Mixture-of-Experts Layer, ICLR 2017
> [2] Fedus et al., Switch Transformers: Scaling to Trillion Parameter Models with Simple and Efficient Sparsity, JMLR 2022
> [3] Schick et al., Toolformer: Language Models Can Teach Themselves to Use Tools, NeurIPS 2023
> [4] Qin et al., ToolLLM: Facilitating Large Language Models to Master 16000+ Real-World APIs, ICLR 2024
> [5] Patil et al., Gorilla: Large Language Model Connected with Massive APIs, arXiv 2023
> [6] Xia et al., Evaluating Mathematical Reasoning Beyond Accuracy, AAAI 2025

---

### Author Response · Authors · 2025-11-24
**Summary of Changes**

We sincerely thank all of you for the time and effort invested in reviewing our paper and for providing such thoughtful comments and constructive suggestions. Below, we provide a list of all major changes highlighted in blue in the revised paper.

1. We added prior work on selecting reasoning modes before inference in the section Related Work.
2. We added the detailed explanation of how the computational efficiency is quantified in the subsection Strategy Adapter.
3. We supplemented the necessity of the auxiliary loss in the subsection Strategy Adapter.
4. We refined the description of Deliberative Routing in the subsection Adaptive Routing Policy at Inference.
5. We added more experiments and analyses of strategy effectiveness and routing patterns across difficulty levels and mathematical subjects.
6. We added sensitivity analysis in the Appendix A.3.
7. We introduced the detailed bad case analysis in the Appendix A.7.

---

### Note · Authors · 2026-01-03

I have read and agree with the venue's withdrawal policy on behalf of myself and my co-authors.